# Early childhood investment impacts social decision-making four decades later

Yi Luo[1], Sébastien Hétu[1,2], Terry Lohrenz [1], Andreas Hula[3], Peter Dayan[4,5], Sharon Landesman Ramey[1], Libbie Sonnier-Netto[1], Jonathan Lisinski[1], Stephen LaConte[1], Tobias Nolte[4,6], Peter Fonagy [6,7], Elham Rahmani[8], P. Read Montague[1,4] & Craig Ramey[1]

Early childhood educational investment produces positive effects on cognitive and non-cognitive skills, health, and socio-economic success. However, the effects of such interventions on social decision-making later in life are unknown. We recalled participants from one of the oldest randomized controlled studies of early childhood investment—the Abecedarian Project (ABC)—to participate in well-validated interactive economic games that probe social norm enforcement and planning. We show that in a repeated-play ultimatum game, ABC participants who received high-quality early interventions strongly reject unequal division of money across players (disadvantageous or advantageous) even at significant cost to themselves. Using a multi-round trust game and computational modeling of social exchange, we show that the same intervention participants also plan further into the future. These findings suggest that high quality early childhood investment can result in long-term changes in social decision-making and promote social norm enforcement in order to reap future benefits.

[1] Virginia Tech Carilion Research Institute, Roanoke, VA 24016, USA. [2] Université de Montréal, Montreal, QC H3C 3J7, Canada. [3] Austrian Institute of Technology, 1210 Vienna, Austria. [4] Wellcome Trust Centre for Neuroimaging, University College London, 12 Queen Square, London WC1E 6BT, UK. [5] Gatsby Computational Neuroscience Unit, University College London, London WC1E 6BT, UK. [6] Anna Freud National Centre for Children and Families, 21 Maresfield Gardens, London NW3 5SD, UK. [7] Research Department of Clinical, Educational and Health Psychology, University College London, Gower Street, London WC1E 6BT, UK. [8] Psychiatry Department, Virginia Tech Carilion School of Medicine, Roanoke, VA 24016, USA. These authors contributed equally: Yi Luo, Sébastien Hétu. Correspondence and requests for materials should be addressed to P.R.M. (email: read@vtc.vt.edu)

Early childhood investment improves the development of disadvantaged children through affecting a variety of cognitive and non-cognitive skills, often translating into better outcomes during adulthood[1–3]. The Abecedarian Project (ABC) —one of the world's oldest high-quality experiments of early childhood intervention—enrolled newborns from low-income, multi-risk families in Orange County, North Carolina, between 1972 and 1977, and provided intensive early childhood education intervention from the first few months of life until school entry. Follow-up studies have provided mounting evidence for positive cognitive[4], educational[5], economic[6], and physical health[7] outcomes into adulthood for participants who were exposed to this intervention. However, possible effects of early childhood interventions on social decision-making strategies have not yet been investigated in this population. This is an important issue as certain social decision-making strategies could benefit an individual, including later financial, educational, social, and health outcomes. One such strategy is to choose actions that enforce social norms such as equality. Social norm enforcement, which often entails a cost[8], is thought to be motivated by the fact that it can result in long-term positive effects on cooperation[9] and thus lead to future benefit—outweighing the immediate cost—for the individual. Since the development of social decision-making styles can be traced back to early childhood[10,11], it is therefore essential to investigate if early childhood intervention can impact social decision-making later in adulthood. In the current study, we used two economic games to probe decision-making during social interactions in ABC participants at ages 39–45: the ultimatum game (UG) particularly effective at measuring enforcement of social norms of equality and fairness[12] and the multi-round trust game (MRT) measuring the process of cooperation forming/ rupture through iterated social exchanges[13,14].

In the UG aimed at probing social decision-making related to norm enforcement, one player (Proposer) has to decide how to split a sum of money with another player (Responder). The Responder's acceptance results in each party receiving the allocated money whereas rejection results in both receiving no money[15]. The UG builds a context in which players have to make trade-offs between self-interest and social norms of equality— rejections being a way to punish behaviors that transgress such social norms[16]. A prominent hypothesis for explaining such behavior is the Fehr–Schmidt inequality aversion model, which proposes that people use a utility function that expresses preferences for equality and away from inequality, in both disadvantageous (i.e., Responder gets less than the Proposer) and advantageous (i.e., Responder gets more than the Proposer) situations[17]. The Fehr–Schmidt model is consistent with the fact that rejection of disadvantageous (low) offers has been consistently observed across studies[15,18]. However, behavior towards advantageous offers is more variable[19–21]. A recent study found participants rejected both disadvantageous and advantageous offers more than equal offers when playing the UG as a third party not involved in the distributive outcome. In contrast, when playing as Responders whose own benefit was affected by their choices, these same participants did not reject advantageous offers more than equal offers[22]. This finding suggests that even if individuals aspire to promote and enforce an "equal world", self-interest can often overcome inequality aversion.

The UG focuses on equality, but ignores strategic considerations that arise when repeatedly interacting with the same player. These involve planning over multiple rounds, along with the necessity of characterizing one's partners and modeling the iterative exchange. To examine this, each participant also played an MRT of 10 rounds with the same partner. In each round, one player (Investor) received $20 and had to choose to invest any portion of it. This amount of money was tripled and sent to the other player (Trustee) who

decided how much of it to repay the Investor[13]. A recent model of preference and mental states has provided a quantitative way to model behavior during this task[23]. This model assumes that players compute the long-run utilities of the available options to guide decisions[24] and it allows us to investigate both immediate reactions such as inequality aversion and their capacity to plan ahead— operationalized by the model's planning horizon. The planning horizon is important since failing to plan ahead during a social exchange can lead to a rupture in cooperation[14].

In this study we showcase how combining economic games with sophisticated computational models of behavior can provide sensitive indicators to assess the long-term effects—more than 40 years later—of early educational interventions on social decision-making. With this ecological approach, as we show next, such intervention can result in long-term changes in social decision-making and promote social norm enforcement possibly for the consideration of future benefits.

## Results

**ABC project overview.** The ABC project was conducted with children from low-income families in the 1970s (see Supplementary Materials in ref. [7] for details). It was designed to examine the impact of intensive early childhood education on preventing developmental delays and academic failure. One hundred and twenty families were recruited in the ABC Project with Scores of High Risk Index[25] as eligibility criteria (infants with a score higher than 11 were eligible[4]). Enrolled families were paired on High Risk Index scores and then one family from each pair was randomly assigned to either the intervention or the control group. The base sample included 111 children from 109 families that accepted their randomization assignment and accepted to participate (one family with a pair of twins and another with a pair of siblings). Among them, 57 were assigned to the intervention group; the other 54 to the control group. At the preschool stage (from 2 months to 5 years of life), for both intervention and control groups, a standard intervention including nutritional, health care, and family social support services was given, while the intervention group received an extra educational intervention. The educational intervention included cognitive and social stimulation, caregiving, and supervised play throughout a full 8-h day (5 days per week, 50 weeks per year) during the first 5 years, emphasizing language, emotional regulation, and cognitive skill development[26,27]. This intervention employed a curriculum including a series of "educational games" developed by Sparling and Lewis[26]. Each teacher was in charge of child care for three infants, which ensured intensive interactions between children and teachers. In addition, the intervention group was also provided with free primary pediatric care and nutrition, including routine screenings, immunizations, pediatric care staff visits, and laboratory tests. Another randomization was implemented when children entered schools when they were five years old, with about half of the children in the intervention or control group during the preschool stage assigned to another three-year intervention[28,29]. Analyses in the current paper focused on the outcomes for intervention in the preschool phase, irrespective of the assignments at the school-age stage[5].

From the original 111 ABC participants, 78 took part in the current study, 36 (ABC Controls) received basic supports (i.e., nutritional, health care, and family social support services) from birth to age 5, while 42 (ABC Interventions) received these supports along with a 5-year, high-quality educational intervention focusing on cognition and social–emotional development (see Table 1 and Supplementary Table 1 showing attrition was comparable in both groups). We also tested an independent control group of 252 adults (Roanoke Controls) recruited from

**Table 1 Participant retention and attrition**

| | Intervention Female | | Intervention Male | | Control Female | | Control Male | |
|---|---|---|---|---|---|---|---|---|
| | Followed | Attritted | Followed | Attritted | Followed | Attritted | Followed | Attritted |
| Initial enrollment (N) | 28 | 0 | 29 | 0 | 31 | 0 | 23 | 0 |
| Fifth decade (N) | 24 | 4 | 26 | 3 | 26 | 5 | 19 | 4 |
| Behavior analysis (N) | 20 | 8 | 22 | 7 | 21 | 10 | 15 | 8 |

Roanoke, Virginia, who did not receive any controlled intervention in early childhood.

**Stronger norm enforcement in ABC Interventions in UG.** Our participants played 60 rounds of UG as the Responder deciding whether to accept or reject offers about how to split $20 with different Proposers (Fig. 1a). Unknown to the participants, offers were generated by a computer algorithm which selected the offers from three different truncated Gaussian distributions: Low offers (mean = $4), Medium offers (mean = $8), or High offers (mean = $12), each with a standard deviation = $1.5. Participants were randomly assigned to one of two conditioning types: the medium-high–medium (MHM) type and the medium-low–medium (MLM) type. During the first 20 rounds, both types received offers taken from the Medium distribution. During the next 20 rounds, the MHM type received offers from the High distribution while the MLM type received offers taken from the Low distribution. For the last 20 rounds, both types received offers from the Medium distribution (Fig. 1a). All participants were asked to report their emotional reaction towards the current offer from unhappy to happy on a 1–9 scale in 60% of the rounds. To examine social behavior towards both disadvantageous and advantageous inequality, the MHM type was the main focus of our analysis on rejection rate and emotion rating, since advantageous offers (higher than $10) were rarely displayed in MLM type (see Supplementary Figs. 1 and 2, Supplementary Table 2 and Supplementary Note 1 for results for MLM type).

Using Group (ABC Intervention vs. ABC Control vs. Roanoke Control) × Equality (Disadvantageous Unequal vs. Equal vs. Advantageous Unequal) analyses of covariance (ANCOVA) with Gender as the covariate and Bonferroni-corrected post hoc $t$-tests, we found that ABC Controls who underwent the MHM conditioning type rejected disadvantageous offers (mean ± s.e. m., 42.7 ± 8.5% of all disadvantageous offers for one participant) more than equal offers (1.0 ± 1.0%), $p < 0.001$, but did not reject advantageous offers (4.5 ± 2.5%) more than equal offers (1.0 ± 1.0%), $p = 1.000$. This pattern of response is in line with previous work that reported behavior driven by self-interest in the face of advantageous offers (i.e., low rejection rates)[22,30] and very similar to the pattern we observed in Roanoke Controls. In stark contrast, along with rejecting disadvantageous offers (48.3 ± 7.2%), ABC Interventions (in MHM type) rejected advantageous offers (43.4 ± 8.5%) more than equal offers (5.0 ± 2.8%), $p$'s < 0.001. This difference in rejection pattern was confirmed by a significant Group × Equality interaction, $F(3,258) = 13.464$, $p < 0.001$, $\eta^2_p = 0.140$ (Fig. 2b), and the fact that ABC Interventions rejected advantageous offers more than ABC Controls ($p < 0.001$) and Roanoke Controls ($p < 0.001$). A significant Group × Offer Size interaction was also found, $F(5,411) = 8.229$, $p < 0.001$, $\eta^2_p = 0.090$ (see Supplementary Note 2 for details). In keeping with this, only the ABC Interventions showed a "V shape" rejection rate pattern, with rejection increasing as a function of inequality, regardless of whether the inequality was personally advantageous or disadvantageous (Fig. 2c and Table 2). Since rejecting offers in the UG is akin to punishing the Proposer[31], this rejection pattern

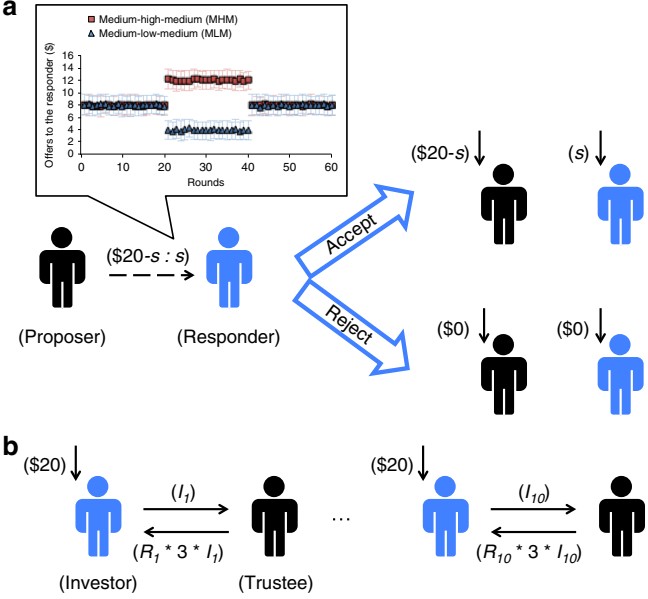

**Fig. 1** The ultimatum game (UG) and multi-round trust game (MRT) in the current study. **a** Procedure for UG comprising of 60 rounds, with Mean and SD of offer size in each round across Medium-Low-Medium (MLM) or Medium-High-Medium (MHM) conditioning type. Each participant was told he/she was the Responder in this game who decided to accept or reject the offer (s) from different Proposers in each round. Offers were sampled from one of the three Gaussian distributions: low offers (mean $4, SD $1.5); medium offers (mean $8, SD $1.5); and high offers (mean $12, SD $1.5). **b** Procedure for MRT comprising 10 consecutive rounds. Participants were told that they, as the Investor, were playing with the same Trustee across the whole game. In each round, the participant received $20 and decided how much of it to send to the Trustee. This amount of money (I) received by the Trustee was tripled (3*I) and any portion of it was then repaid to the investor (R*3*I). SD standard deviation

can be considered a strong social signal aimed at enforcing equality during exchanges.

Taking advantage of behavioral modeling, we also estimated individual sensitivity to advantageous and disadvantageous inequality in the UG using the Fehr–Schmidt inequality aversion utility function[17] where the utility of an offer is discounted by "envy" (unwillingness to accept disadvantageous offer) and "guilt" (unwillingness to accept advantageous offers). In line with our rejection rate results, an ANCOVA with Gender as the covariate and Bonferroni-corrected post hoc $t$-tests found that while ABC Interventions had similar envy coefficients as ABC Controls ($p = 1.000$), they had higher guilt coefficients than both ABC Controls ($p < 0.001$) and Roanoke Controls ($p < 0.001$) (Fig. 2d), suggesting heightened sensitivity to advantageous inequality (see Supplementary Note 3 for details). Furthermore, the similar envy and guilt coefficients in ABC Interventions

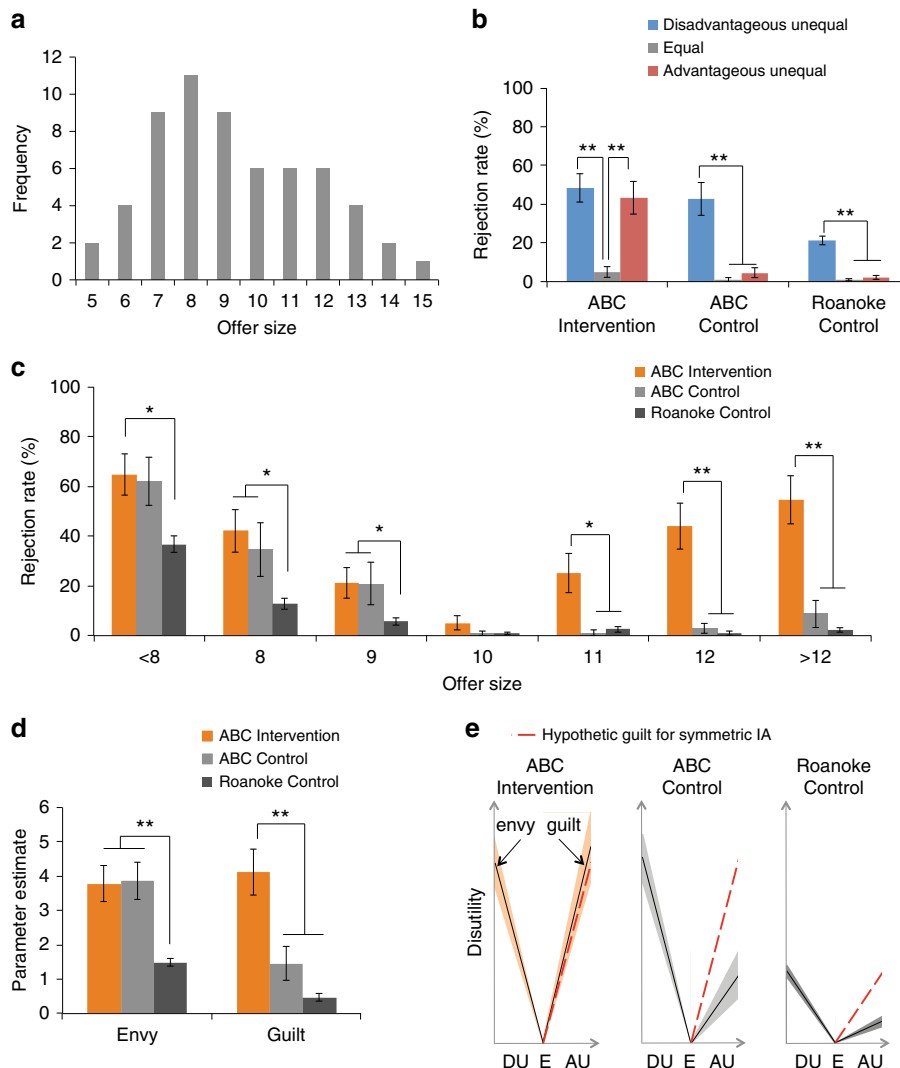

**Fig. 2** Offer distribution, rejection rates, and model-based parameters in the ultimatum game. **a** Distribution of offer size for Medium-High-Medium (MHM) conditioning type. The frequency is the average occurrence for each offer size across participants in MHM. **b** Rejection rates for MHM grouped by level of equality. All groups had higher rejection rates for disadvantageous offers than equal offers. Advantageous offers were not rejected more than equal offers in ABC Control and Roanoke Control, while ABC Interventions rejected advantageous offers more than equal offers. **c** Rejection rates for MHM grouped by offer size. Only ABC Interventions increased rejection rates as a function of inequality, regardless of them being personally advantageous or disadvantageous, presenting a "V shape" pattern. **d** Parameter estimates from the behavioral modeling using a Fehr–Schmidt inequality aversion model. Both ABC Interventions and ABC Controls have a higher level of envy (unwillingness to accept unequal offers which are disadvantageous to the participant) than Roanoke Controls. The ABC Interventions had a higher guilt (unwillingness to accept unequal offers which are advantageous to the participant) than ABC Controls and Roanoke Controls. **e** The horizontal axis presents different levels of equality: DU disadvantageous unequal, E equal, AU advantageous unequal. The vertical axis presents the disutility defined by the inequality aversion (IA) model (i.e., sensitivity × inequality). The slope of each line presents the sensitivity for inequality aversion (IA; envy for DU and guilt for AU). A steeper slope corresponds to higher inequality aversion. Compared with the control groups, ABC Interventions presented a much more symmetric IA pattern (i.e., the same level of envy and guilt). Shaded areas are bounded by mean ± s.e.m. *$p < 0.05$, **$p < 0.001$ (post hoc $t$-test $p$-values). Error bar represents s.e.m.

suggest that, as a group, they displayed a unique symmetric disutility for advantageous and disadvantageous offers which is in accordance with their "V shape" rejection pattern (Fig. 2e). Finally, when asked to report their feelings towards the offers, only ABC Interventions rated equal offers as more pleasant than both advantageous and disadvantageous offers ($p$'s < 0.001), consistent with symmetric sensitivity to inequality and the "V shape" rejection pattern (Fig. 3 and Table 3). Overall, these results suggest that an early childhood intervention program can profoundly alter sensitivity to norm deviation in adulthood and promote social norm enforcement.

**ABC Interventions planned further into future in MRT**. In the MRT, two players—an Investor and a Trustee—engage in 10 rounds of economic exchange game. In the current study, each participant played 10 consecutive rounds as the Investor (Fig. 1b). A model based on interactive partially observable Markov decision processes[32] (iPOMDP) was used to characterize preference and mental states during social interaction within the MRT. The model assumes that players compute the long-run utilities (called $Q$-values) of the available options to guide decisions[24]. A self-consistency condition for the $Q$-values over successive rounds is prescribed by the Bellman equation[33]. Based on extensive vali-

**Table 2 Rejection rates between offer sizes in Medium-High-Medium conditioning type**

| Offer size | Rejection rate (mean ± s.e.m.) | Comparison | | | Difference |
|---|---|---|---|---|---|
| *ABC Intervention* | | | | | |
| offers < 8 | 64.9 ± 8.4% | > | offers = 8 | | $p = 0.003$ |
| offers = 8 | 42.2 ± 8.6% | > | offers = 9 | | $p < 0.001$ |
| offers = 9 | 21.3 ± 6.2% | > | offers = 10 | | $p = 0.003$ |
| offers = 10 | 5.0 ± 2.8% | < | offers = 11 | | $p < 0.001$ |
| offers = 11 | 25.2 ± 7.9% | < | offers = 12 | | $p < 0.001$ |
| offers = 12 | 44.1 ± 9.4% | < | offers > 12 | | $p = 0.002$ |
| offers > 12 | 54.5 ± 9.7% | – | – | | – |
| *ABC Control* | | | | | |
| offers < 8 | 62.0 ± 9.6% | > | offers = 8 | | $p = 0.014$ |
| offers = 8 | 34.7 ± 10.9% | n.s. | offers = 9 | | $p = 0.118$ |
| offers = 9 | 20.8 ± 8.6% | > | offers = 10 | | $p = 0.006$ |
| offers = 10 | 1.0 ± 1.0% | n.s. | offers = 11 | | $p = 1.000$ |
| offers = 11 | 1.1 ± 1.1% | n.s. | offers = 12 | | $p = 1.000$ |
| offers = 12 | 3.0 ± 2.1% | n.s. | offers > 12 | | $p = 1.000$ |
| offers > 12 | 8.8 ± 5.5% | – | – | | – |
| *Roanoke Control* | | | | | |
| offers < 8 | 36.7 ± 3.4% | > | offers = 8 | | $p < 0.001$ |
| offers = 8 | 12.8 ± 2.2% | > | offers = 9 | | $p = 0.001$ |
| offers = 9 | 5.6 ± 1.3% | n.s. | offers = 10 | | $p = 0.233$ |
| offers = 10 | 1.0 ± 0.6% | n.s. | offers = 11 | | $p = 1.000$ |
| offers = 11 | 2.5 ± 1.2% | n.s. | offers = 12 | | $p = 1.000$ |
| offers = 12 | 0.9 ± 0.8% | n.s. | offers > 12 | | $p = 1.000$ |
| offers > 12 | 2.3 ± 1.0% | – | – | | – |

*s.e.m.* standard error of the mean
*n.s.* means the difference was not significant
*P*-values were Bonferroni corrected

dation[23], this model has two structural characteristics. First, since the value of a player's action depends on the future decisions of the partner, players are assumed to develop a model of their partners—which assumes players have distinct levels of theory of mind and follow a cognitive hierarchy theory[34]. Specifically, a player of type *k* assumes that the partner has type *k*-1. Second, players are assumed to model *Q*-values only a certain number (the planning horizon) of rounds into the future, substituting default values thereafter (Fig. 4a).

We first assessed possible group difference with one-way analyses of variance on overall behavior (fractional investments) and performance (total earnings) across the MRT. Average fractional investments were similar across groups, $F(2,327) = 2.488$, $p = 0.085$, $\eta^2_p = 0.015$ (ABC Interventions, $50.6 \pm 3.2\%$; ABC Controls, $45.9 \pm 3.2\%$; Roanoke Controls, $54.5 \pm 1.5\%$). The total earnings were also not different among groups, $F(2,327) = 2.443$, $p = 0.088$, $\eta^2_p = 0.015$ (ABC Interventions, $215.67 \pm 6.02$; ABC Controls, $210.5 \pm 7.60$; Roanoke Controls, $225.07 \pm 2.68$).

However, using model-based analysis, we were able to highlight group differences in decision-making strategies, with ABC Interventions having a higher level of planning horizon compared to Roanoke Controls and a lower level of ToM compared to ABC Controls. To be more specific, with independent-samples Kruskal–Wallis *H*-test and the follow-up two-tailed Mann–Whitney *U*-tests, we found a significant main effect of group for planning horizon, $H(2) = 6.849$, $p = 0.033$, with ABC Interventions ($2.45 \pm 0.17$) having a higher level of planning horizon than Roanoke Controls ($2.01 \pm 0.07$, Bonferroni corrected $p = 0.027$) but not significantly different from ABC Controls ($2.08 \pm 0.20$, Bonferroni-corrected $p = 0.269$) (Fig. 4b). This result suggests that social decision-making in ABC Interventions, compared to Roanoke Controls, seems to be particularly influenced by future and long-term outcomes. The main effect of group was also found to be significant for theory of mind (ToM), $H(2) = 6.701$, $p = 0.035$, with ABC Controls ($2.83 \pm 0.24$) having a higher level of ToM than ABC Interventions ($2.05 \pm 0.24$, Bonferroni-corrected $p = 0.047$) but not different

from Roanoke Controls ($2.62 \pm 0.09$, Bonferroni corrected $p = 0.974$), suggesting that ABC Controls might have adopted a different social decision-making strategy (i.e., modeling the other) compared to ABC Interventions' strategy (i.e., planning ahead). See Supplementary Note 4 for details about other parameters in this model.

## Discussion

The ABC project and its follow-up studies have shown that early childhood intervention can result in short- and long-term effects including important and substantial cognitive, health, and educational benefits[5,7,29]. In line with these results, now over 40 years after, ABC Interventions in the current study, compared to ABC Controls, reported higher levels of "very close" relationships with their parents (85.7% vs. 58.3%, $p < 0.001$), higher levels of educational attainment (97.6% vs. 75.0% completed high school, $p < 0.001$ and had four times the rate of college graduation), and more of them had saving accounts (92.9% vs. 66.7%, $p < 0.001$) (see Supplementary Table 3 and Supplementary Note 5 for midlife demographic information). These real-life outcomes might profit greatly from the ability to attend to social norms, establish and maintain positive social interactions, and plan into the future, which are dynamic and complex processes related to social decision-making.

The current study revealed a boost in social norm enforcement behaviors and sensitivity to social norm violation in middle-age adults from the ABC project who received a high-quality educational intervention during the first 5 years of their lives. Playing the role of the Responder in an UG, these individuals rejected unequal offers more than equal offers regardless of whether the split was disadvantageous or advantageous to them—displaying symmetric inequality aversion. Since rejecting the offers of the Proposers in the UG is akin to punishing the Proposer[31], this rejection pattern can be considered as a strong social signal aimed at enforcing equality during social exchanges. Indeed, a study investigating the motivations of rejecting advantageous offers by analyzing verbal data during the discussion between pairs of players in the UG[21] found that most participants claimed to enforce the norm of equality when they rejected offers that were advantageous to them.

Our results revealed that ABC Interventions rejected advantageous offers more than the two control groups while both ABC Interventions and ABC Controls rejected disadvantageous offers more than Roanoke Controls. This result for disadvantageous offers could be related to differences between the ABC and Roanoke samples: ABC groups had much more experience with psychological testing than the Roanoke group, ABC participants are African-American while the Roanoke Controls are more racially diverse, and the samples grew up in different states (North Carolina vs. Virginia). However, the amount of testing in the follow-up studies of the ABC project was equivalent between ABC Interventions and ABC Controls, and importantly, neither the intervention nor the follow-up tests involved economics games such as the UG (or MRT) for either ABC group. Hence the higher rejection rate on advantageous offers compared to the two control groups and the unique symmetric inequality aversion of ABC Intervention is more likely driven by the effects of the educational intervention during the first five years of their lives. We note that these differences are based on a relatively small sample which might raise the possibility of it being a false positive. However, the amount of convergent evidence, including the unique V shape rejection rate pattern of ABC Interventions, the group difference on model-based parameters and the corresponding emotion rating patterns, suggests that our result of a

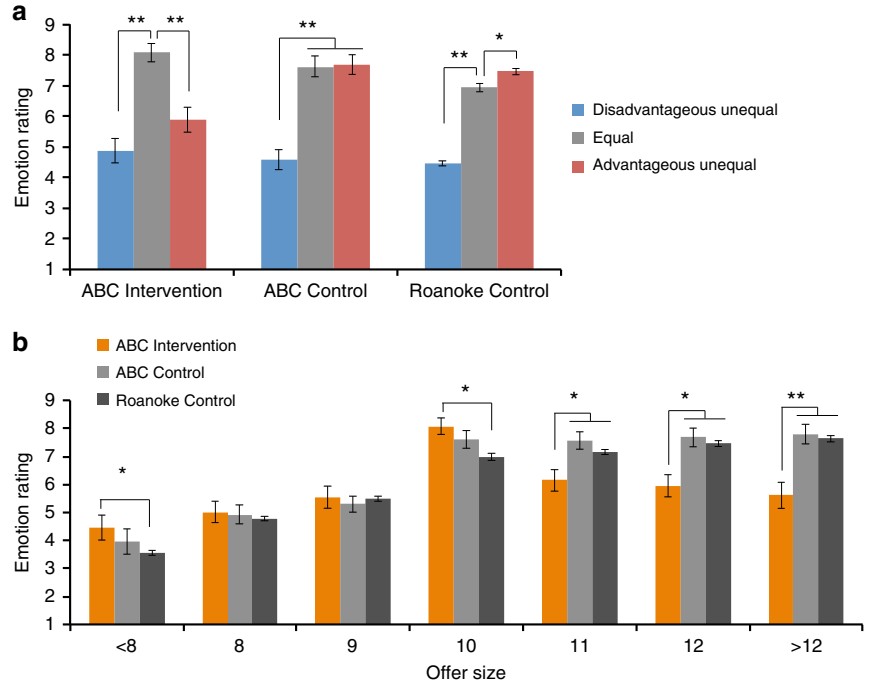

**Fig. 3** Emotion rating across 60 rounds in Medium-High-Medium (MHM) type in the ultimatum game. **a** Emotion rating for MHM grouped by level of equality. ABC Interventions rated equal offers as more pleasant than both disadvantageous and advantageous offers. ABC Controls rated disadvantageous offers as less pleasant than equal offers but did not report different feelings about equal and advantageous offers. Roanoke Controls rated disadvantageous offers as less pleasant than equal offers but advantageous offers as more pleasant than equal offers. **b** Emotion rating across 60 rounds in MHM grouped by offer size. ABC Interventions rated equal offer as more pleasant than each disadvantageous offer as well as than each advantageous unequal offer, while ABC Controls decreased emotion rating for more disadvantageous offers but reported no difference between equal and advantageous offers. Compared with ABC Controls and Roanoke Controls, ABC Interventions rated each advantageous offer as significantly less pleasant. *$p < 0.05$, **$p < 0.001$ (post hoc $t$-test $p$-values). Error bar represents s.e.m.

**Table 3 Emotion ratings between offer sizes in Medium-High-Medium conditioning type**

| Offer size | Emotion ratings (mean ± s.e.m.) | Comparison | | Difference |
|---|---|---|---|---|
| *ABC Intervention* | | | | |
| offers < 8 | 4.5 ± 0.4 | < | offers = 8 | $p = 0.027$ |
| offers = 8 | 5.0 ± 0.4 | < | offers = 9 | $p = 0.004$ |
| offers = 9 | 5.5 ± 0.4 | < | offers = 10 | $p < 0.001$ |
| offers = 10 | 8.1 ± 0.3 | > | offers = 11 | $p < 0.001$ |
| offers = 11 | 6.1 ± 0.4 | n.s. | offers = 12 | $p = 1.000$ |
| offers = 12 | 5.9 ± 0.4 | n.s. | offers > 12 | $p = 1.000$ |
| offers > 12 | 5.6 ± 0.5 | – | – | – |
| *ABC Control* | | | | |
| offers < 8 | 4.0 ± 0.5 | < | offers = 8 | $p < 0.001$ |
| offers = 8 | 4.9 ± 0.3 | n.s. | offers = 9 | $p = 0.874$ |
| offers = 9 | 5.3 ± 0.3 | < | offers = 10 | $p < 0.001$ |
| offers = 10 | 7.6 ± 0.3 | n.s. | offers = 11 | $p = 1.000$ |
| offers = 11 | 7.6 ± 0.3 | n.s. | offers = 12 | $p = 1.000$ |
| offers = 12 | 7.7 ± 0.3 | n.s. | offers > 12 | $p = 1.000$ |
| offers > 12 | 7.8 ± 0.4 | – | – | – |
| *Roanoke Control* | | | | |
| offers < 8 | 3.6 ± 0.1 | < | offers = 8 | $p < 0.001$ |
| offers = 8 | 4.8 ± 0.1 | < | offers = 9 | $p < 0.001$ |
| offers = 9 | 5.5 ± 0.1 | < | offers = 10 | $p < 0.001$ |
| offers = 10 | 7.0 ± 0.1 | n.s. | offers = 11 | $p = 1.000$ |
| offers = 11 | 7.2 ± 0.1 | < | offers = 12 | $p < 0.001$ |
| offers = 12 | 7.5 ± 0.1 | n.s. | offers > 12 | $p = 1.000$ |
| offers > 12 | 7.6 ± 0.1 | – | – | – |

s.e.m. standard error of the mean
n.s. means the difference was not significant
*P*-values were Bonferroni corrected

difference in equality-based decision-making between ABC Interventions and Controls is a robust finding.

When resources are divided unequally, people typically exhibit norm enforcement in which they punish the person responsible for the unequal split, even at a cost[35]. However, self-interest often overweighs the willingness to enforce this social norm[22]—unequal splits that are advantageous are less punished than disadvantageous ones[30]. Indeed, contrary to ABC Interventions who showed symmetric inequality aversion, our two control groups were unwilling to forgo the possibility of enjoying the personal benefits of advantageous offers which they rarely rejected. This important difference suggests that early educational interventions —in this case, for children from deeply impoverished backgrounds—can contribute to stronger norm enforcement during social exchanges in adulthood. Our results are in line with recent findings from another early childhood education program showing that preschool education made children more egalitarian at 7–8 years old[36]. Importantly, our results demonstrate that such changes can extend into adulthood—many decades after the intervention. It was recently shown that disadvantaged children whose mothers are more prosocial (measured by higher levels of altruism, trust, and other-regarding preference) had a higher increase in prosociality after receiving interventions that enriched their social environment[37]. Our results, showing that early investment in children can influence social decision-making (i.e., higher other-regarding and inequality-aversive in terms of monetary distribution) during adulthood thus suggest that facilitating prosociality behavior through early childhood investments could have a cascading effect—through intergenerational transmission—by potentializing the impacts of subsequent interventions in future generations. It is notable that the effect found in the current study may be relatively restricted in terms of its translation to everyday behaviors. It would be interesting to look in future work at commonplace measures that are closely related to inequality (such as charitable giving) in order to see if our

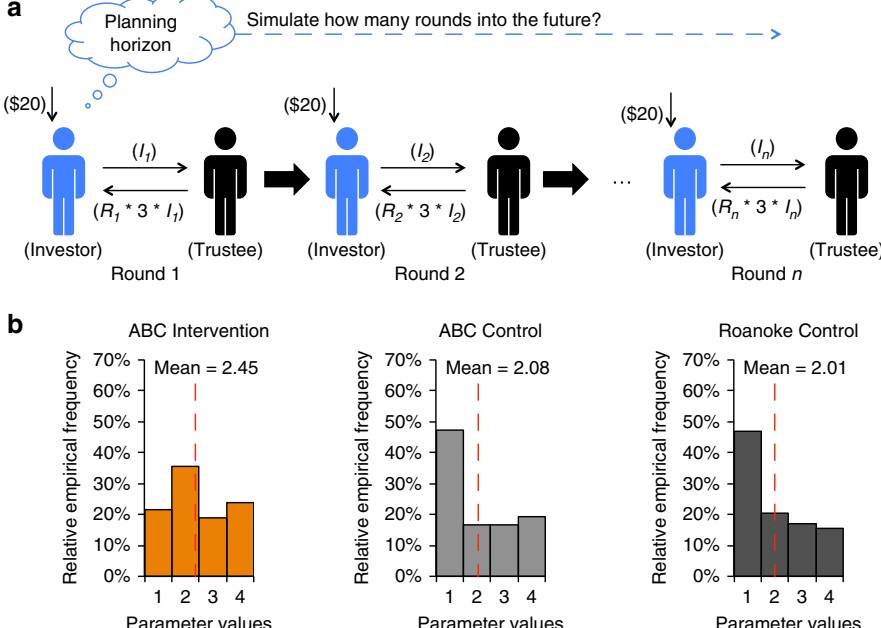

**Fig. 4** The Planning horizon in the multi-round trust game (MRT). **a** In each round, the Investor (played by the participant; indicated in blue) received $20 and decided how much of it to send to the Trustee (played by a computer algorithm; indicated in black). The amount (*I*) the Investor sent was tripled and delivered to the Trustee, who decided what fraction (*R*) of this total (3\**I*) to send back to the Investor. The Investor ended up with 20-*I*+*R*\*3\**I*; the Trustee with (1-*R*)\*3\**I*. Planning horizon quantifies how many steps of the future interactions the participant took into account when assessing the results of his/her investment during the MRT. **b** Distribution of the future planning capacity parameter (planning horizon) in ABC Intervention, ABC Control, and Roanoke Control group, respectively

laboratory findings are paralleled by social decision-making in their daily life.

In the MRT, additionally, although no differences were found on overall investment and performance between groups, using computational modeling we were able to highlight differences in social decision-making strategies. On the one hand, we observed that ABC Interventions planned further into the future in the MRT compared to Roanoke Controls. On the other hand, the higher ToM parameter for ABC Controls indicated that this group utilized more mentalization steps than ABC Interventions during this game. These findings suggest that the similar overall behavior of the two ABC groups might be motivated by different social decision-making strategies. It is possible that ABC Interventions focused more on future social interactions while ABC Controls took other's mental states into more consideration, potentially indicative of a preference to reap short-term benefits. These findings also reveal the advantage of our approach which provides information about the underlying mechanism (or strategy) of social decision-making, and thus go beyond the study of the outcome (or overall behavior) of this process. It would be important for future studies to examine the role of these underlying strategies (e.g., planning horizon and ToM) on social decision-making using experimental designs. It is notable that these process-oriented results found in the current study could be related to actual outcomes of social decision-making (e.g., income or other economic outcomes;social well-being). Future work on this would add ecological validity of these process-oriented findings.

The striking high rejection of advantageous unequal offers in the UG in the current study is rare in large-scale societies (but see in refs. [21,38]). However, previous work on participants from Western cultures found that when people played the UG as a third party whose personal benefit would not be affected by their decisions, they also displayed a symmetric inequality aversion[22],

which is very similar to what we find here in ABC Interventions. This suggests that such symmetric inequality aversion is indeed valued by some in Western cultures. Importantly, studies[22] including our own have also shown that this social norm is less enforced when self-interest comes into play, which may explain why it is rarely observed. The rejection of advantageous offers also occurs in some small-scale societies where the convention is to repay an unsolicited gift—high value offers are sometimes rejected when the Responder does not want to be indebted to the Proposer[39]. It raises the possibility that the ABC Interventions in our study might have a stronger belief in reciprocity so that they were more alert to potential cost or repayment caused by advantageous offers in the future, instead of current payoffs. It is just one possible hypothesis; in the future, it should be tested using more elaborate methods because of the obvious difference between these small-scale societies and the culture of our participants.

The longer planning horizon of the ABC Interventions estimated from the MRT suggests their unique norm enforcement behavior during the UG may be motivated by such anticipated future personal costs but also potential social benefits. Consistent with this interpretation, equality enforcement by punishing unequal offers was shown to be positively correlated with altruistic behaviors across cultures[40]. Indeed, behavior towards equality and fairness despite short-term cost to self has been proposed as important in stabilizing long-term cooperation[9,41]. Furthermore, ensuring that others comply with social norms can impact others' behaviors to create a more cooperative social environment[8], which can produce future societal benefits.

Our approach based on quantitatively prescribed economic games provides a quantitative indicator evaluating possible social benefits of early childhood investment programs. By having participants actively interact with others in social contexts, we provide a more ecologically valid way to measure outcomes on

social decision-making compared to self-report questionnaires. Indeed, our findings contribute unique knowledge about long-term correlates of receiving a high-quality early education: changes in social decision-making highlighted by higher social norm enforcing behavior probably motivated by anticipated future benefits related to social cooperation. We should hasten to add that our findings seem to reveal the effects of the educational intervention in the first five years of the ABC participants' life. However, this does not necessarily exclude the possibility that these differences are related to other factors that resulted from the intervention and that occurred during the four decades that followed it. Indeed, since our measures were taken at a single time point (over 40 years after the intervention), known changes on health[7], educational level[5], social connections as well as other factors induced by the intervention may have also played a role in shaping the participants' social decision-making pattern. Our own analyses did not detect an association between adult educational attainment, for example, and decision-making behavior in this sample. It would be valuable, nonetheless, to further investigate if and how early childhood interventions or any aspects of them can give rise to differences in later experiences that in turn influence social decision-making behaviors. By investing in the early education of highly vulnerable children—with a program that underscores positive adult–child interactions, explicitly teaches about cause-and-effect, permits active learning and early decision-making opportunities, and promotes increasingly complex social cooperation—children realize a brighter future, becoming healthier[7], more productive[3], and as our results show, stronger promoters of the norms on which our society is built.

## Methods

**Participants.** As mentioned above, the base sample included 111 children. About four decades after their enrollment, 16 participants attrited with 9 deceased (3 from the intervention group, 6 from the control group), 2 in prison (1 from the intervention group, 1 from the control group), and 5 withdrawal (3 from the intervention group, 2 from the control group; see Table 1 for the pattern of attrition). Among the remaining 95 participants (50 from the intervention group, 45 from the control group), 78 participants, now in their 40s, took part in the current study (participation rate: 82%). Among them, 42 received the standard intervention and the extra educational intervention (ABC Intervention group) while 36 were in the control group during early childhood (ABC Control group) and only received the standard intervention. Several midlife demographic information were also assessed as part of this stage of the ABC project using various self-reported questionnaires (see Supplementary Table 3 and Supplementary Note 5 for midlife demographicinformation).

Additionally, 252 adult participants in Roanoke, Virginia also took part in this study. These participants did not receive any controlled intervention during their childhood (Roanoke Control group).

All participants gave written consent to participate in the experiments, and all procedures were performed in accordance with the Institutional Review Board of the Virginia Tech Carillion Research Institute. Data about gender and age are presented in Supplementary Table 5.

**Task procedure.** Participants' social behaviors were assessed using the UG and MRT. Stimuli were presented and responses were collected using NEMO (Human Neuroimaging Laboratory, Virginia Tech Carilion Research Institute). The order of the two games was randomized across participants. Details for non-deception and incentivization in the two games are provided in Supplementary Note 6.

In the UG, one player (Proposer) has to decide how to split a $20 endowment between himself and another player (Responder). The Responder has to decide whether to accept or reject the offer. If the offer is accepted, both players get the proposed amounts. However, if the offer is rejected, both players get $0. Participants played 60 rounds as the Responder and were informed that the offers in each round were from different proposers. Abecedarian participants and Roanoke participants were randomly assigned to one of two conditioning type: the medium-high-medium type (MHM) and the medium-low-medium type (MLM) (17 in MLM and 25 in MHM for ABC Interventions; 21 in MLM and 15 in MHM for ABC Controls; 122 in MLM and 130 in MHM for Roanoke Controls). During the first 20 rounds, participants assigned to both types received offers taken from the Medium distribution (Mean = $8, SD = $1.5). During the next 20 rounds, participants assigned to the MHM type received offers from the High distribution (Mean = $12, SD = $1.5) while those assigned to MLM type received offers taken

from the Low distribution (Mean = $4, SD = $1.5). For the last 20 rounds, participants from both types received offers from the Medium distribution (Fig. 1a). In three of five rounds after having made their decisions, participants were also asked to rate their emotion towards the current offer on a 1–9 scale, ranging from unhappy to happy using emoticons adapted from the self-assessment manikin[42]. Participants were informed that they would be paid according to outcomes in one randomly selected round and were encouraged to treat each round as the selected round. The timeline of one round of the UG is illustrated in Supplementary Fig. 3a.

In the MRT, in each round, one player (Investor) received $20 and had to choose to invest any portion of it. This amount of money was tripled and sent to the other player (Trustee) who decided how much of it to repay the Investor. Each participant played 10 consecutive rounds as the Investor in the MRT with the same partner. Unknown to the participants, the Trustee's responses in the MRT were generated using a k-nearest neighbors sampling algorithm on known responses from real players as described in ref. [14]. The timeline of one round of the MRT is illustrated in Supplementary Fig. 3b.

**Statistics.** Statistics were implemented using SPSS software (IBM SPSS Statistics Version 21.0, IBM Corp.). For all analyses, the significance level was set at 0.05 and Greenhouse–Geisser correction non-sphericity was used when appropriate. Post hoc comparisons were evaluated using two-tailed pairwise tests with Bonferroni correction. Partial eta-squared ($\eta^2_p$) values were provided to demonstrate effect size where appropriate[43].

**Rejection rate and emotion rating in UG.** Our analyses focus on the participants' social behavior towards inequality. Specifically, we tested the interaction between equality and treatment on rejection rate by a (3 × 3) Group (ABC Intervention vs. ABC Control vs. Roanoke Control) × Equality (Disadvantageous Unequal (offers < 10) vs. Equal (offers = 10) vs. Advantageous Unequal (offers > 10)) analyses of covariance (ANCOVA) with Gender as the covariate in the MHM type to take account of the gender unbalance within MHM type. As offers higher than $10 were rarely displayed for in the MLM type, a (3 × 2) Group (ABC Intervention vs. ABC Control vs. Roanoke Control) × Equality (Disadvantageous Unequal (offers < 10) vs. Equal (offers =10)) ANCOVA with Gender as the covariate was used for the MLM type. Gender was included as a covariate in the analyses to control for the difference in gender composition in the three groups.

Further, the rejection rates for each offer were calculated for each participant—making it possible to precisely describe social behavior from highly disadvantageous, to highly advantageous inequality. Because of the different distribution of offers between MHM and MLM (Fig. 2a and Supplementary Fig. 1a) resulting in small number of offers for certain amounts depending on the conditioning type, different offer sizes were evaluated for MHM and MLM. For each participant with MHM type, offers lower than $8 and offers higher than $12 were respectively pooled together. For each participant with MLM type, offers lower than $5 were pooled together and offers higher than $10—rarely displayed—were not included into the behavioral data analysis in MLM type. For each conditioning type, differences in rejection rates were analyzed using a (3 × 7) Group × Offer Size ANCOVA with Gender as the covariate to take account of the gender unbalance within MLM type.

Differences in emotion ratings were analyzed using a similar approach: a Group × Equality ANCOVA and an Offer Size × Group ANCOVA with Gender as the covariate for each type of conditioning.

**Model-based analyses for behaviors in UG.** We assumed that the participants' behavior could be modeled by their aversion to offers that deviate from equality and fitted each participant's behaviors to a Fehr–Schmidt inequality aversion model (FS model)[17]. As previous studies using the UG have shown that people have internal norms (expectations on money allocation) which can be updated based on the history of offers[16,44,45], we also fitted the behavioral data to two types of adaptation models, a Bayesian observer model[44] and a Rescorla–Wagner model[33,46] to test if they outperformed the FS model.

In the FS inequality aversion model, the utility of each offer at each round was represented by the Fehr–Schmidt inequality aversion utility function.

$$U(s_i) = s_i - \alpha \max\{10 - s_i, 0\} - \beta \max\{s_i - 10, 0\}. \quad (1)$$

Here, $U(s_i)$ represents the utility of the offer $s_i$ at round $i$. This value is discounted by the difference between the amount allocated (offer) to the Responder ($s_i$) and an even split ($10). The disutility associated with inequality is controlled by two parameters: $\alpha$ or "envy" ($\alpha \in [0,10]$) which represents the participant's unwillingness of the participant to accept unequal offers disadvantageous to him/her; $\beta$ or "guilt" ($\beta \in [0,10]$) which represents his/her unwillingness to accept unequal offers advantageous to him/her.

The probability of accepting each offer was modeled using a softmax function:

$$p_{\text{accept}} = \frac{e^{u * \gamma}}{1 + e^{u * \gamma}}. \quad (2)$$

Here, $\gamma$ is the softmax inverse temperature parameter where the lower $\gamma$ is, the more diffuse and variable the choices are ($\gamma \in [0,1]$).

Under the Bayesian observer model[44] (BO model), we assumed that each participant believed that the offers were sampled from a Gaussian distribution with uncertain mean and variance, and performed Bayesian update after receiving a new offer. Specifically, each participant was assumed to have a prior on the distribution of the offers ($s$), with mean $\mu$ and variance $\sigma^2$, denoted as $s \sim N(\mu, \sigma^2)$. Since $\mu$ and variance $\sigma^2$ were mixed together, the prior of offers ($s$) was assumed as $p(\mu, \sigma^2)$. The prior was updated following Bayes' rule once the participant received a new offer. The posterior was given by

$$p(\mu\, \sigma^2 | s_i) = \frac{p(s|\mu, \sigma^2) p(\mu | \sigma^2)}{p(s_i)}. \tag{3}$$

For convenience we assumed a conjugate prior of $\mu$ and $\sigma^2$:

$$p(\mu, \sigma^2) = p(\mu, | \sigma^2)\, p(\sigma^2), \tag{4}$$

with

$$p(\mu | \sigma^2) = \mathrm{Normal}(\hat{\mu}, \hat{\sigma}^2 / k) \tag{5}$$

$$p(\sigma^2) = \mathrm{Inv} - \chi^2(\nu, \hat{\sigma}^2). \tag{6}$$

We set the initial value of the hyperparameters $k$, $\nu$ and $\hat{\sigma}^2$ as

$$k_0 = 4, \ \nu_0 = 10, \ \hat{\sigma}_0^2 = 4$$

Two variations of the BO models were tested. The first assumed equality as a fixed initial norm for all participants, $\hat{\mu}_0 = 10$. The second assumed that the initial norm could vary between participants, hence $\hat{\mu}_0$ was individually fitted using each participant's responses ($\hat{\mu}_0 \in [0, 20]$).

After receiving $s_i$, at round $i$, these values were updated as

$$k_i = k_{i-1} + 1, \nu_i = \nu_{i-1} + 1, \tag{7}$$

$$\hat{\mu}_i = \hat{\mu}_{i-1} + \frac{1}{k_i}(s_i - \hat{\mu}_{i-1}), \tag{8}$$

$$\nu_i \hat{\sigma}_i^2 = \nu_{i-1}\hat{\sigma}_{i-1}^2 + \frac{k_{i-1}}{k_i}(s_i - \hat{\mu}_{i-1})^2. \tag{9}$$

We define the prevailing norm as $\mu_{i-1}$ at round $i$, and the utility of the offer is given by

$$U(s_i) = s_i - \alpha \max\{\mu_{i-1} - s_i, 0\} - \beta \max\{s_i - \mu_{i-1}, 0\}. \tag{10}$$

Here, $\alpha$ represents the unwillingness of the participant to accept offers lower than his/her norm ($\alpha \in [0,10]$). $\beta$ represents the unwillingness to accept offers higher than him/her norm ($\beta \in [0,10]$).

The probability of accepting each offer was

$$p_{\mathrm{accept}} = \frac{e^{u*\gamma}}{1 + e^{u*\gamma}}, \tag{2}$$

where $\gamma \in [0,1]$.

The Rescorla–Wagner (RW) model assumed that each participant had internal norms which were updated by the RW rule:[33,46]

$$x_i = x_{i-1} + \varepsilon(s_i - x_{i-1}). \tag{11}$$

Here $x_i$ represents the norm at round $i$ and $\varepsilon$ is the norm adaptation rate ($\varepsilon \in [0,1]$), which represents the extent to which the norm was influenced by the difference (i.e., norm prediction error) between the current offer $s_i$ and the preceding norm $x_{i-1}$. A low $\varepsilon$ indicates a lower impact of the norm prediction error on norm updating whereas a high $\varepsilon$ indicates a high impact. Similar to the BO model, two variations of the RW model were tested based on the initial norm $x_0$: a fixed initial norm based on equality ($x_0 = 10$) and variable initial norms across participants. The utility of an offer at round $i$ is given by

$$U(s_i) = s_i - \alpha \max\{x_{i-1} - s_i, 0\} - \beta \max\{s_i - x_{i-1}, 0\}. \tag{12}$$

Similar to BO model, $\alpha$ here represents the unwillingness of the participant to accept offers lower than his/her norms ($\alpha \in [0,10]$). $\beta$ represents the unwillingness to accept offers higher than him/her norms ($\beta \in [0,10]$).

The probability of accepting each offer was

$$p_{\mathrm{accept}} = \frac{e^{u*\gamma}}{1 + e^{u*\gamma}}, \tag{2}$$

where $\gamma \in [0,1]$.

All models were then fitted to the behavioral data individually, which estimated the values of $\alpha$, $\beta$, $\gamma$, and $\hat{\mu}_0$ or $x_0$ for variable starting norm models for each subject by maximizing the log likelihood of choices over 60 trials. Then model comparison was implemented by calculating the Bayesian information criterion score (BIC) for each model for each participant. The model with the lowest mean BIC is considered the winning model since it has the maximal model evidence (Supplementary Table 4). The estimated parameters from the winning model were compared among the three groups of participants with an ANCOVA with Gender as a covariate. Post hoc comparisons were evaluated using Bonferroni correction. The ranges of the free parameters in models presented above were based on previous work[44]. We tested other ranges that resulted in slightly worse model fitting and did not significantly affect the results presented here.

**Fractional investment in MRT**. The fractional investment sent from the participant (Investor) in each round was calculated as the amount of investment in each round divided by the resource available to the investor in the current round (i.e., $20). The group difference on the average of investment across 10 rounds was tested with a one-way analysis of variance.

**Total earning in MRT**. The earning of the participant (Investor) in each round was calculated as the amount from the $20 kept by the Investor plus the amount of repayment sent from the Trustee. The group difference on the sum of earnings across 10 rounds was tested with a one-way analysis of variance.

**Model-based analyses for behaviors in MRT**. The foundation of players' payoff assessment was also based on the Fehr–Schmidt inequality aversion model[17], but the envy term of the equation was omitted for the MRT. To distinguish the guilt parameter in the MRT from the one in the UG, the guilt parameter in the MRT is called inequality aversion, which quantifies the tendency to try and reach a fair outcome with values of {0, 0.4, 1}.

This model of MRT includes six other parameters: (1) planning horizon, which quantifies number of steps to likely plan ahead with values of {1, 2, 3, 4}; (2) theory of mind (ToM), which quantifies the number of mentalization steps with values of {0, 2, 4}; (3) inverse temperature, which quantifies the randomness of the choice preference with values of {1/4, 1/3, 1/2, 1}; (4) risk aversion, which quantifies the value of money kept over money potentially gained with values of {0.4, 0.6, 0.8, 1.0, 1.2, 1.4, 1.6, 1.8}; (5) irritability, which quantifies tendency to retaliate to repayments worse than expected with values of {0, 0.25, 0.5, 0.75, 1.0}; and (6) irritability Belief, which quantifies the initial belief of likelihood of the partner being irritable with values of {0, 1, 2, 3, 4}. The values for the parameters were selected based on previous work[23]. The whole collection of parameters that best characterize an individual player are determined by maximizing the likelihood of their choices (over a grid of possible values). See ref. [23] for a detailed description of the model. Since these seven parameters from the model were ordinal variables, the tests of group effects for each parameter were conducted with independent-samples Kruskal–Wallis $H$ tests. Post hoc comparisons were evaluated using two-tailed Mann-Whitney $U$-tests with Bonferroni correction.

**Code availability**. The code used to analyze data in the current study is available from the corresponding author on request.

**Reporting Summary**. Further information on research design is available in the Nature Research Reporting Summary linked to this article.

## Data availability
The datasets generated and/or analyzed during the current study are available from the corresponding author on request. A reporting summary for this article is available as a Supplementary Information.

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

## Acknowledgements

We thank Laura Bateman and Carrie Bynum for their assistance. Our extreme gratitude also goes out to the families and individuals who have participated in this work over these many years. This work was funded by a Principal Research Fellowship from the Wellcome Trust (to P.R.M.), Virginia Tech (to P.R.M. and S.L.R.), and the FRQ-S and CIHR (to S.H.). P.F. is receipt of a National Institute for Health Research (NIHR) Senior Investigator Award (NF-SI-0514-10157). P.F. was in part supported by the NIHR Collaboration for Leadership in Applied Health Research and Care (CLAHRC) North Thames at Barts Health NHS Trust. The views expressed are those of the authors and not necessarily those of the NHS, the NIHR, or the Department of Health. P.D. was funded by the Gatsby Charitable Foundation; he is currently at Max Planck Institute for Biological Cybernetics, Tuebingen, Germany.

## Author contributions

Y.L., S.H., T.L., and P.R.M. designed the ultimatum game and the multi-round trust game experiment; C.R. and S.R. designed the ABC experiment and follow-up; Y.L., S.H., S.L.R., C.R., L.S. and P.R.M. performed research; Y.L., S.H., T.L., A.H., P.D., J.L., T.N., P. F., and P.R.M analyzed data; Y.L., S.H., T.L., A.H., P.D., S.L.R., J.L., S.L., C.R., T.N., P.F., E.R. and P.R.M. discussed the results; Y.L., S.H., T.L. and P.R.M. wrote the manuscript. A.H., P.D., S.L., T.N., S.L.R. and C.R. provided important comments and suggestions that significantly contributed to the manuscript.

## Additional information

**Competing interests:** The authors declare no competing interests.

