## [Peer Review File · Nature Communications]

Reviewers' Comments:

Reviewer #1:

None

Reviewer #2:

Remarks to the Author:

In this paper, the authors recall participants in the ABC study of early childhood interventions to test their decisions on two economic games, the Ultimatum game (UG) and a repeated trust game (MRT). They compared subjects who had received the standard intervention (ABC control) to those who received the standard intervention plus intensive educational intervention (ABC Intervention) to an age matched control group from the same geographic area (Roanoke Control). In the UG, individuals in both ABC groups were more likely to refuse disadvantageous unequal offers than those in the Roanoke Control, but those in the ABC Intervention were more likely to refuse advantageous unequal offers than those in either Control group. All three groups' overall earnings and decisions were the same in the MRT, but model analysis of behavioral data indicated that ABC Intervention participants were planning ahead more than the control groups, and that ABC Control groups were using theory of mind more than the other two groups. The authors conclude that the educational intervention led to improvements in social decision-making skills that in particular led to a focus on equity and equality (as evidenced by an aversion to advantageous inequity despite the personal gains that they would receive) and an enhanced ability to plan as compared to either of the other two groups.

The procedures run in this study are sound and I have no concerns. They used a reasonable population control (the Roanoke Control) and the UG and MRT are standard procedures used in experimental economics to study social decision-making. I do have some concerns about the interpretation that I think that the authors need to consider further.

My primary concern is that I feel like the effects of the educational intervention are overstated. First of all, there are no objective differences in outcome in one of the two games; all three groups showed the same investment patterns and earnings in the MRT, they simply differed in the (inferred) intentions in how they played the game, based on the model. This begs the question of how critical these differences are if they don't lead to changes in outcome. Second, in the UG, both ABC groups were the same – and different from the Roanoke Control – in the disadvantageous situation, which leads me to a second concern, which is that the key result – the difference in advantageous inequity responses in the ABC Intervention group – could be simply because this group of individuals are highly experienced with psychology and economic testing situations. We know they interacted with the experimenters more, as part of the intervention, but how much testing has this group had, and has it been more than the ABC Control individuals?

Second, the authors interpret the ABC Intervention group's responses as being more like the results from cross-cultural studies that show occasional refusals of overbenefitting responses, particularly in cultures with strong gift giving cultures and traditions. I agree that they are similar to each other, but how do we interpret this? The authors argue that they are sensitive to fairness norms, but another interpretation is that these individuals are really atypical in that they look more like Lamalera whale hunters than their own culture. What does that mean?

In addition, how do we respond to the fact that even more traditional societies do not respond to the same degree as modern Western societies in the disadvantageous situation? This has led to the argument that modern Western societies are "WEIRD", which could indicate that what the educational intervention really did was teach these kids how to become super-"WEIRD" – that is, to internalize WEIRD social norms. That's not necessarily counter to what the authors argue (i.e., that they learn to value equity) but the authors present it as if valuing equity were somehow objectively better and the natural state of things that we should aim for. This section needs more discussion of how these kids developed these views and what it really means.

Third, the authors interpret these data as showing that the educational intervention influenced subjects' social decision making, along with a host of other factors that have been shown in previous work. But how do we know that it's the educational intervention that influenced social decision-making, and not the fact that they are better educated and have savings accounts and plans for the future? Of course they all presumably ultimately come from the same original cause, but we can't say after 40 years that it was the intervention and not some other factor that resulted from it. This makes a difference because we need to know what's really important – this particular intervention or some aspect of it, such as being better educated or having a more hopeful future (i.e., having money in savings). That needs a more nuanced discussion.

Line 120 – “combining”, not “combing”

Reviewer #3:

Remarks to the Author:

The manuscript by Prof Montague and colleagues reports evidence that early childhood investment impacts social decision-making 40 years later: Participants in the intervention treatment are (1) significantly more inequality averse with respect to advantageous inequality than a control group, and (2) plan further into the future in social interactions.

From my perspective, the study is highly relevant as it is very important to assess the long-term consequences of early childhood investment on variables such as social decision making. This task is very difficult due to the long time horizon between intervention and data collection (in the current study 40 years). The study builds on a very interesting data source – the Abecedarian Project (ABC) – which includes both an intervention group and a control group and the authors are able to approach approximately 40 participants in each group from the original sample of about 55 participants per group.

I find the study very interesting and well-written, however, I have concerns regarding the robustness of the observed effects, which are outlined below.

(a) Forty participants per treatment typically appear as a good rule of thumb, if effect sizes are unknown (as it is probably the case in the current manuscript). To derive conclusion (1), however, the authors split their sample and end up with a comparison of 25 subjects from the intervention group vs. 15 participants of control group. Underpowered studies are more likely to generate false positive results (Button et al. 2013) and calls this this result into question.

(b) The second result appears also difficult from a statistical point of view: First, no statistical differences in overall behavior and performance were observed. Only in the model-based analysis, differences were identified. Of course, testing average behavior and the model-based parameters increases the number of total tests and thereby the chance of detecting results by chance. Second, to me, only the comparison between ABC Controls vs. ABC Interventions is really informative to derive meaningful conclusions on the intervention. The comparison ABC Controls vs. ABC Interventions, however, does not reach the significance threshold of $p < 0.05$ for the planning horizon calling this result also into question.

Minor points:

The authors apply economic games to study social preferences. However, they do not follow the main rules developed within this field, which are non-deception and incentivization. I am aware of the differences among scientific fields, but I think the authors should at least discuss that they did not follow these rules and explain why.

It would be helpful for the reader, if the authors would report exact p-values at least within a certain range such as $p < 0.001$ to $0.250 < p$, as p-values contain important information.

I think it would be good to make the data and code for analysis publicly available on some online platform such as OSF.io. This facilitates data sharing.

References

Button, K. S., Ioannidis, J. P., Mokrysz, C., Nosek, B. A., Flint, J., Robinson, E. S., & Munafò, M. R. (2013). Power failure: why small sample size undermines the reliability of neuroscience. *Nature Reviews Neuroscience*, 14(5), 365.

Response to Reviewers

We thank the reviewers for their comments and constructive suggestions. We have addressed them all, and believe the manuscript has improved significantly. Changes to the manuscript are indicated in red text.

Please note that, after a further data check, we found one duplicate subject (from Roanoke Control in the medium-low-medium [MLM] type). We excluded data from this duplicate subject and reran the analyses on the corrected sample (42 ABC Interventions, 36 ABC Controls, and 252 Roanoke Controls). Since the duplicate subject was from the Roanoke Control group in the MLM type, the main results focusing on the MHM type were not changed after excluding this subject. Results regarding to MLM type remained the same except for some minor changes in statistical values which have been revised and are highlighted in the manuscript and in the supplementary information.

Reviewers' comments:

Reviewer #2 (Remarks to the Author):

In this paper, the authors recall participants in the ABC study of early childhood interventions to test their decisions on two economic games, the Ultimatum game (UG) and a repeated trust game (MRT). They compared subjects who had received the standard intervention (ABC control) to those who received the standard intervention plus intensive educational intervention (ABC Intervention) to an age matched control group from the same geographic area (Roanoke Control). In the UG, individuals in both ABC groups were more likely to refuse disadvantageous unequal offers than those in the Roanoke Control, but those in the ABC Intervention were more likely to refuse advantageous unequal offers than those in either Control group. All three groups' overall earnings and decisions were the same in the MRT, but model analysis of behavioral data indicated that ABC Intervention participants were planning ahead more than the control groups, and that ABC Control groups were using theory of mind more than the other two groups. The authors conclude that the educational intervention led to improvements in social decision-making skills that in particular led to a focus on equity and equality (as evidenced by an aversion to advantageous inequity despite the personal gains that they would receive) and an enhanced ability to plan as compared to either of the other two groups.

The procedures run in this study are sound and I have no concerns. They used a reasonable population control (the Roanoke Control) and the UG and MRT are standard procedures used in experimental economics to study social decision-making. I do have some concerns about the interpretation that I think that the authors need to consider further.

My primary concern is that I feel like the effects of the educational intervention are overstated. First of all, there are no objective differences in outcome in one of the

two games; all three groups showed the same investment patterns and earnings in the MRT, they simply differed in the (inferred) intentions in how they played the game, based on the model. This begs the question of how critical these differences are if they don't lead to changes in outcome.

Response: We appreciate the reviewer's concern about our statements regarding the effects of the education intervention.

Although no significant difference in the observed overall behaviors was found in the MRT, striking behavioral effects were found in the UG. It is these latter results that are the main findings of our manuscript.

Indeed, we show that while both control groups had very low rejection rates for advantageous offers (not different than rejection rate for equal offers), participants who received the education intervention rejected these high offers significantly more than equal offers (also see response to Reviewer #3's first comment where we provide more evidence based on individual behavior inspection and other measures). When looking more closely at this behavioral effect by breaking down each offer size we also show that only the ABC Interventions increased their rejection rates as a function of inequality, regardless of whether the split was disadvantageous or advantageous to them. This finding is supported by a further analysis on the group difference by offer size which revealed a V-shape rejection rate only for ABC Interventions, and by the model-based result that ABC Interventions had a much higher guilt parameter (i.e., aversion to advantageous inequality) compared to both control groups. Importantly, we also show that the intervention had an effect on another behavior related to the UG: only ABC Interventions rated both advantageous and disadvantageous offers as less pleasant than equal offers.

This remarkable finding drives and is directly related to the most important conclusion of the current study that high quality early childhood investment can result in long-term changes in social decision-making and promote social norm enforcement. We acknowledge that this effect may be relatively restricted in terms on how it translates to everyday behaviors. Indeed, these are based on several different variables that can sometime include inequality (e.g., giving to charities). We add this point in the manuscript and call for future studies testing the effect of educational intervention on real-life decision-making.

Discussion (Page 16) *“It is notable that the effect found in the current study may be relatively restricted in terms of its translation to everyday behaviors. It would be interesting to look in future work at commonplace measures that are closely related to inequality (such as charitable giving) in order to see if our laboratory findings are paralleled by social decision-making in their daily life.”*

We completely agree with the reviewer that understanding WHICH decisions are made is critical but we also think that understanding HOW decisions are made is also critical. Even if the observed overall behaviors of different groups are the same, the underlying

decision processes leading to this behavior can be different. For example, when choosing between an apple and a Pop-tart at the supermarket, two individuals may choose the apple, but knowing that one chose it because the apple is healthier while the other one because the apple is cheaper gives us a much richer understanding of their decision-making processes.

This is where a model-based approach can be very useful. It opens the door to studying the underlying processes and individual characteristics (such as inequality aversion, planning horizon, and theory of mind in our tasks) that bring forth the observed behavior/outcome. In regard to the MRT, our approach using ecological tasks in combination with computational modeling shows differences in model-based parameters suggesting that the similar investment behavior of the three groups in the current study were possibly driven by different processes.

We agree that these results are inferences that are only based on our model. Therefore, we modified the manuscript to call for future study to test the hypotheses about the underlying mechanism during social decision-making by experiments targeting specific processes and strategies. We also have endorsed the value in relating these decision-making differences to other behavioral outcomes in this and other study sample(s), such as their income and other economic outcomes and their personal-social well-being. This would add ecological validity of these process-oriented findings.

Discussion (Page 16) *“It would be important for future studies to examine the role of these underlying strategies (e.g., planning horizon and ToM) on social decision-making using experimental designs. It is notable that these effects found in the current study may be related to other behavioral outcomes in this and other study sample(s), such as their income and other economic outcomes and their personal-social well-being. Future work on this would add ecological validity of these process-oriented findings.”*

Second, in the UG, both ABC groups were the same – and different from the Roanoke Control – in the disadvantageous situation, which leads me to a second concern, which is that the key result – the difference in advantageous inequity responses in the ABC Intervention group – could be simply because this group of individuals are highly experienced with psychology and economic testing situations. We know they interacted with the experimenters more, as part of the intervention, but how much testing has this group had, and has it been more than the ABC Control individuals?

Response: The reviewer is right to point out that in the disadvantageous situation, both ABC groups had a similar rejection rate, a rate that was indeed higher than the Roanoke Controls. The reviewer is also right to point out that the ABC groups had much more experience with (or exposure to) psychological testing than the Roanoke group. However, we want to inform the reviewer that ABC Interventions did not have more contact or experience with psychology and economic testing situations than the ABC Controls: contact from infancy until this age 40 assessment was identical with the treated and

untreated controls in ABC and all assessments were conducted by individuals who had no other contact with the subjects, their families, or their teachers.

The study design may be limited by other differences between the Roanoke Controls and the ABC participants, since ABC subjects were African-American while the Roanoke controls are more racially diverse and the sample grew up in different states (North Carolina vs. Virginia). We now address these limits in the manuscript.

Discussion (Page 15) *“Our results revealed that ABC Interventions rejected advantageous offers more than the two control groups while both ABC Interventions and ABC Controls rejected disadvantageous offers more than Roanoke Controls. This result for disadvantageous offers could be related to differences between the ABC and Roanoke samples: ABC groups had much more experience with psychological testing than the Roanoke group, ethnicity (ABC participants are African-American while the Roanoke Controls are more racially diverse) and the samples grew up in different states (North Carolina vs. Virginia).”*

Importantly, we strongly believe that our key result – the difference in advantageous inequity responses in ABC Intervention vs. Control group – cannot only be explained by the amount of testing the groups had undergone. This is for several reasons.

First, the educational intervention included cognitive and social stimulation, responsive caregiving, and supervised play during the first 5 years, emphasizing language, emotional regulation, and cognitive skill development but did not include economic testing similar to the ones used in the current study.

Second, ABC participants have never been tested on these economic tasks (UG and MRT) or any similar ones in follow-up studies after the intervention. Indeed, at no time since the enrollment have economic games or decision-making tasks been included in assessment. Further, the last behavioral assessments of the ABC subjects occurred 10 years prior to this assessment and then 9 years earlier. It seems improbable that an hour or two of assessment one or two decades earlier could impact the current behavior of ABCs.

Third, the amount of testing in the follow-up studies of the ABC project was equivalent between ABC Interventions and ABC Controls but we still found important differences between them when we tested them with our tasks (especially in the UG).

In other words, since the ABC Intervention group did not have more psychology and economic testing experiences than the ABC Control group, this suggests that the unique symmetric inequality aversion of ABC Intervention is more likely driven by the educational intervention during the first five years of their lives.

Discussion (Page 15) *“However, the amount of testing in the follow-up studies of the ABC project was equivalent between ABC Interventions and ABC Controls, and importantly, neither the intervention nor the follow-up tests involved economics games such as the UG (or MRT) for either ABC group. Hence the higher rejection rate on*

advantageous offers compared to the two control groups and the unique symmetric inequality aversion of ABC Intervention is more likely driven by the effects of the educational intervention during the first five years of their lives.”

Second, the authors interpret the ABC Intervention group’s responses as being more like the results from cross-cultural studies that show occasional refusals of overbenefitting responses, particularly in cultures with strong gift giving cultures and traditions. I agree that they are similar to each other, but how do we interpret this? The authors argue that they are sensitive to fairness norms, but another interpretation is that these individuals are really atypical in that they look more like Lamalera whale hunters than their own culture. What does that mean?

Response: We appreciate the reviewer’s concern based on our interpretation of ABC Intervention group’s responses. We would like to clarify that here we used previous cross-cultural findings in other societies that displayed a similar behavior pattern as the ABC Interventions in an effort to understand the reason of their unique high rejection rate for advantageous unequal offers. Although we believe that interpretations established for these other small-scale societies can provide a useful explanation for the mechanism underlying the symmetric inequality aversion in the ABC Interventions—consideration for reciprocity and future benefit/cost—these hypotheses need to be tested using more elaborate means in the future. We have highlighted this point by modifying the manuscript.

We also agree with the reviewer that ABC Intervention participants seem to be atypical in the sense that their behavior towards advantageous inequality is rare in western societies (their own). However, this does not necessarily mean that their inequality aversion or even overall social functioning is more like those of people from small-scale societies than those from their own culture. Indeed, the rejection of advantageous offers, though rare in large-scale societies, have been documented in Russia¹ and China². Milder versions of this phenomenon have also been observed in western cultures^{3,4}. More importantly, as reported in a previous study, when participants from western culture played the UG as a third party whose personal benefit would not be affected by their decisions, they also displayed a symmetric inequality aversion⁵, which is very similar to what we find here in ABC Interventions. This suggests that such symmetric inequality aversion is indeed valued in the western culture but less enforced when self-interest comes to play (when people are playing as the Responder who can receive more money by accepting advantageous offers).

We have revised our Discussion accordingly to avoid confusion.

Discussion (Page 16 - 17) *“The striking high rejection of advantageous unequal offers in the UG in the current study is rare in large-scale societies (but see in^{1,2}). However, previous work on participants from Western cultures found that when people played the UG as a third party whose personal benefit would not be affected by their decisions, they*

also displayed a symmetric inequality aversion⁵, which is very similar to what we find here in ABC Interventions. This suggests that such symmetric inequality aversion is indeed valued by some in Western cultures. Importantly, studies⁵ including our own have also shown that this social norm is less enforced when self-interest comes into play which may explain why it is rarely observed. The rejection of advantageous offers also occurs in some small-scale societies where the convention is to repay an unsolicited gift—high value offers are sometimes rejected when the Responder does not want to be indebted to the Proposer⁶. It suggests that the ABC Interventions in our study might have a stronger belief in reciprocity so that they were more alert to potential cost or repayment caused by advantageous offers in the future, instead of current payoffs. It is just one possible hypothesis; in the future, it should be tested using more elaborate methods because of the obvious difference between these small-scale societies and the culture of our participants. The longer planning horizon of the ABC Interventions estimated from the MRT suggests their unique norm enforcement behavior during the UG may be motivated by such anticipated future personal costs but also potential social benefits. Consistent with this interpretation, behavior towards equality and fairness despite short-term cost to self has been proposed as important in stabilizing long-term cooperation^{7,8}. It has also been found that equality enforcement by punishing unequal offers is positively correlated with altruistic behaviors across cultures⁹. Furthermore, ensuring that others comply with social norms can impact others' behaviors to create a more cooperative social environment¹⁰, which can produce future societal benefits.”

In addition, how do we respond to the fact that even more traditional societies do not respond to the same degree as modern Western societies in the disadvantageous situation? This has led to the argument that modern Western societies are “WEIRD”, which could indicate that what the educational intervention really did was teach these kids how to become super-“WEIRD” – that is, to internalize WEIRD social norms. That’s not necessarily counter to what the authors argue (i.e., that they learn to value equity) but the authors present it as if valuing equity were somehow objectively better and the natural state of things that we should aim for. This section needs more discussion of how these kids developed these views and what it really means.

Response: We appreciate the reviewer’s comments and suggestion. We do not suggest that differences in norm enforcement are related to a more (or less) sophisticated or WEIRD way of making inequality-based decision-making. Similar or different patterns of behavior are probably more in line with differences arising from different experiences (learning) which can trace back to culture-based learning or in our case childhood educational intervention.

We try to remain agnostic to the “quality” of this behavior. However, the previous finding that people from WEIRD societies tend to show a preference for equality when there is no monetary incentive⁵ suggests that deep down, the symmetric inequality aversion is the norm that the society they are part of would like to promote. It has also

been found that equality enforcement is positively correlated with altruistic behaviors across cultures⁹. These findings lead us to propose that they are able to make inequality-based social decision-making—to support social norms for the benefit of all—in a less self-interested way.

We revised some statements in the Discussion to be more value-agnostic about ABC Interventions' enhanced inequality sensitivity and norm enforcement.

Discussion (Page 15 - 16) *“Importantly, our results demonstrate that such **changes** can extend into adulthood—many decades after the intervention...Our results, showing that early investment in children can **positively** influence social decision-making (i.e., higher other-regarding and inequality-averse in terms of monetary distribution) during adulthood...”*

Discussion (Page 17) *“Indeed, our findings contribute unique knowledge about long-term correlates of receiving a high-quality early education: **changes in** social decision-making—higher social norm enforcing behavior probably motivated by anticipated future benefits related to social cooperation.”*

We also added some discussion about the symmetric aversion found in the interventions means (as in response to reviewer #2's previous comment) and what might have caused it in the Discussion (as in response to reviewer #2's third comment).

Third, the authors interpret these data as showing that the educational intervention influenced subjects' social decision-making, along with a host of other factors that have been shown in previous work. But how do we know that it's the educational intervention that influenced social decision-making, and not the fact that they are better educated and have savings accounts and plans for the future? Of course they all presumably ultimately come from the same original cause, but we can't say after 40 years that it was the intervention and not some other factor that resulted from it. This makes a difference because we need to know what's really important – this particular intervention or some aspect of it, such as being better educated or having a more hopeful future (i.e., having money in savings). That needs a more nuanced discussion.

Response: We agree with the reviewer that we cannot be certain that after 40 years it was the intervention and not other factors that resulted from the early intervention that contributed to the observed effects on social decision-making. This is a debate about assigning primary attribution or direct effects of the intervention versus secondary or indirect effects. We agree this is impossible to disaggregate. Since our measures were taken at one time point (about 40 years after the start of the intervention), other changes induced by the intervention (including better education, income, and health outcomes) reasonably may be involved in shaping the participants' social decision-making pattern.

To test if being better educated by adulthood played a role in shaping different behavior patterns in the UG, we ran Spearman correlation tests and did not find any significant

correlation between education level and any UG measurement (rejection rates to disadvantageous offer, $r_s(76) = -0.051, p = 0.657$; rejection rates to advantageous offers, $r_s(69) = -0.093, p = 0.439$; emotion ratings to disadvantageous offer, $r_s(76) = 0.138, p = 0.228$; emotion ratings to advantageous offer, $r_s(67) = 0.134, p = 0.272$; envy, $r_s(76) = -0.012, p = 0.914$; guilt, $r_s(76) = -0.127, p = 0.269$). Spearman rank-correlation coefficients were also calculated to test the role of having saving accounts in shaping the behaviors in the UG. We did not find any significant correlation between the possession of saving accounts and any UG measurement (rejection rates to disadvantageous offer, $r_s(76) = -0.076, p = 0.509$; rejection rates to advantageous offers, $r_s(69) = -0.005, p = 0.965$; emotion ratings to disadvantageous offer, $r_s(76) = -0.022, p = 0.851$; emotion ratings to advantageous offer, $r_s(67) = 0.028, p = 0.820$; envy, $r_s(76) = -0.022, p = 0.845$; guilt, $r_s(76) = -0.052, p = 0.649$). Based on these additional results we suggest that the educational intervention is more likely to be the driving influence for detected decision-making patterns in the UG than the education level and possession of saving accounts. However, unfortunately, data generated by the current study could not specifically answer whether any other outcomes (e.g., health) resulting from the educational intervention caused the difference. We addressed this point in the last paragraph of the Discussion by nuancing our interpretations and highlighting that this question is an important direction for future work.

Discussion (Page 17) *“We should hasten to add that our findings seem to reveal the effects of the educational intervention in the first five years of the ABC participants’ life. However, this does not necessarily exclude the possibility that these differences are related to other factors that resulted from the intervention and that occurred during the four decades that followed it. Indeed, since our measures were taken at a single time point (over 40 years after the intervention), known changes on health¹¹, educational level¹², social connections as well as other factors induced by the intervention may have also played a role in shaping the participants’ social decision-making pattern. Our own analyses did not detect an association between adult educational attainment, for example, and decision-making behavior in this sample. It would be valuable, nonetheless, to further investigate if and how early childhood interventions or any aspects of them can give rise to differences in later experiences that in turn influence social decision-making behaviors.”*

Line 120 – “combining”, not “combing”

Response: We thank the reviewer for pointing out this error. We have corrected in the manuscript accordingly.

(Page 4) *“In this study we showcase how **combining** economic games with sophisticated computational models of behavior...”*

Reviewer #3 (Remarks to the Author)

The manuscript by Prof Montague and colleagues reports evidence that early childhood investment impacts social decision-making 40 years later: Participants in the intervention treatment are (1) significantly more inequality averse with respect to advantageous inequality than a control group, and (2) plan further into the future in social interactions.

From my perspective, the study is highly relevant as it is very important to assess the long-term consequences of early childhood investment on variables such as social decision-making. This task is very difficult due to the long time horizon between intervention and data collection (in the current study 40 years). The study builds on a very interesting data source – the Abecedarian Project (ABC) – which includes both an intervention group and a control group and the authors are able to approach approximately 40 participants in each group from the original sample of about 55 participants per group.

I find the study very interesting and well-written, however, I have concerns regarding the robustness of the observed effects, which are outlined below.

(a) Forty participants per treatment typically appear as a good rule of thumb, if effect sizes are unknown (as it is probably the case in the current manuscript). To derive conclusion (1), however, the authors split their sample and end up with a comparison of 25 subjects from the intervention group vs. 15 participants of control group. Underpowered studies are more likely to generate false positive results (Button et al. 2013) and calls this result into question.

Response: We appreciate the reviewer's concern about the potential underpower issue related to the relatively small sample size of the ABC groups. We agree that ideally adding more participants in the analyses would help to increase the power of the study. However, since this study is done on participants who joined the ABC project more than 40 years ago, it is impossible to recruit more participants from the same background who had the same intervention (educational intervention + standard intervention) or control (only standard intervention). Furthermore, we would like to point out that our key finding—ABC Interventions are significantly more advantageous inequality averse than controls—is supported by the following results in our study and would therefore seem unlikely to be a false positive.

First, even though the sample size is relatively small, the group difference reaches statistical significance. The difference in rejection pattern between groups was confirmed by a significant Group \times Equality interaction, $F(3,258) = 13.464, p < 0.001, \eta^2_p = 0.140$, and the fact that ABC Interventions rejected advantageous offers more than ABC Controls ($p < 0.001$) and Roanoke Controls ($p < 0.001$) carried out by post-hoc tests with a stringent correction for multiple comparisons (Bonferroni).

Second, our result is supported by the pattern of individual participants in the ABC Intervention and the ABC Control groups. As shown in Figure R1a and Figure R1b, the majority of ABC Interventions had a higher rejection rate for advantageous offers than

equal offers, while *all but one* ABC Controls rejected much less advantageous offers than disadvantageous offers and most had a 0 rejection rate for advantageous offers (Figure R1c & Figure R1d). Furthermore, we can appreciate how a majority of participants from the ABC Intervention displayed a somewhat symmetric rejection pattern, while almost none of the ABC Controls did. These observations lend support to our conclusion that there is a genuine effect driven by similar behavioral responses in most of the participants (within each group) rather than the result of outliers or a minority of participants with a large effect.

Third, the difference is supported by convergent evidence based on: a) rejection rates by offer size, b) model-based analysis, and c) the emotion rating in the UG. A further inspection of our main result by offer size revealed that only the ABC Intervention group showed a perfectly V-shape rejection rate pattern (Fig. 2c), with rejection rate increasing as a function of inequality, regardless of whether the inequality was personally advantageous or disadvantageous. In line with our rejection rate results, the model-based analysis in the UG revealed that ABC Interventions had higher guilt coefficients than both ABC Controls and Roanoke Controls (Fig. 2d). Finally, the participants' emotion ratings are in accord with our findings on the rejection rate — ABC Interventions rated their emotion regarding both disadvantageous and advantageous offers as less pleasant than that to equal offers, whereas the two control groups rated personally disadvantageous offers as less pleasant than both equal offers and advantageous offers (Fig. 3). We believe that it would be highly unlikely to find a V-shape behavioral pattern as well as the corresponding model-based parameter and emotion rating patterns by chance.

For all these reasons, although our sample size is relatively small, the amount of converging evidence leads us to strongly believe that our result of a difference in equality-based decision-making between ABC Interventions and Controls is not a “false positive”.

However, we appreciate the reviewer's concern about the relatively small sample and the possibility of a false positive, and have pointed it out in the manuscript.

Discussion (Page 15) *“We note that these differences are based on a relatively small sample which might raise the possibility of it being a false positive. However, the amount of convergent evidence, including the unique V shape rejection rate pattern of ABC Interventions, the group difference on model-based parameters and the corresponding emotion rating patterns, suggests that our result of a difference in equality-based decision-making between ABC Interventions and Controls is a robust finding.”*

Figure R1. Rejection rate by level of equality for each participant in the ABC Intervention group (a & b) and in the ABC Control group (c & d). Each line stands for one individual. Plots for each group were split into two figures for better visual display. (a) Rejection rates by level of equality for 13 participants in the ABC Intervention group. (b) Rejection rates by level of equality for the rest 12 participants in the ABC Intervention group. (c) Rejection rates by level of equality for 8 participants in the ABC Control group. (d) Rejection rates by level of equality for the rest 7 participants in the ABC Control group.

(b) The second result appears also difficult from a statistical point of view: First, no statistical differences in overall behavior and performance were observed. Only in the model-based analysis, differences were identified. Of course, testing average behavior and the model-based parameters increases the number of total tests and thereby the chance of detecting results by chance.

Response: We agree with the reviewer that testing both overall behaviour and model-based parameters increases the number of tests and the chance of false positives. However, we suggest that by this approach, we were able to study the underlying processes and individual characteristics (such as inequality aversion, planning horizon,

and theory of mind in our tasks) that bring forth the observed behaviors. We believe that the advantages* of adding these model-based measures are important enough to test them even if adding them increases the chance of detecting “false-positives”. Importantly, we want to point out that we did control for multiple testing on model-based parameters using a conservative Bonferroni correction.

*Please refer to our response to Reviewer #2’s first comment for our argument about the advantages of model-based approaches (We completely agree with the reviewer that understanding WHICH decisions are made is critical but we also think that understanding HOW decisions are made is also critical...)

Second, to me, only the comparison between ABC Controls vs. ABC Interventions is really informative to derive meaningful conclusions on the intervention. The comparison ABC Controls vs. ABC Interventions, however, does not reach the significance threshold of $p < 0.05$ for the planning horizon calling this result also into question.

Response: We agree with the reviewer that the comparison between ABC Interventions and ABC Controls is the most informative way to derive meaningful conclusions on the intervention. As shown in **Figure 4** in the manuscript, we see that ABC Interventions have a clearly different distribution of the Planning horizon from ABC Controls. A large number of participants (almost 50%) in the ABC Control group, similar to those in the Roanoke Control group, had a Planning horizon of 1 step, whereas more participants (almost 40%) in the ABC Intervention group had a Planning horizon of 2 steps. This observation does suggest that ABC Controls are on average different from ABC Interventions and similar to Roanoke Controls. According to our statistical results, ABC Interventions had a longer Planning horizon than the Roanoke Controls (Bonferroni corrected $p = 0.027$) but not significantly different from ABC Controls (Bonferroni corrected $p = 0.269$). It is possible that since the number of participants in the ABC Control group ($N = 36$) is relatively small compared to the number in Roanoke Control ($N = 252$), the difference did not reach the significance threshold of $p < 0.05$. We agree that this non-significant difference does not allow us to assert a strong conclusion about the effects of the intervention on Planning horizon.

We still consider that the pattern we observe here (high Planning horizon in ABC participants) may provide useful information to help understand the unique findings in ABC Interventions for advantageous offers in the UG—possibly enforcing the norm for future considerations. This interpretation is also supported by previous work proposing a relation between future benefit/cost considerations and rejection behavior in the UG⁶.

In agreement with the reviewer’s suggestion, we are now more cautious when interpreting the result about the planning horizon. Instead of claiming that the intervention induced different capability of planning ahead, we suggest that the

consideration for the future as a possible explanation for ABC Interventions' enhanced inequality sensitivity and norm enforcement.

(Page 13) *“Indeed, we found a significant main effect of group for planning horizon, $H(2) = 6.849$, $p = 0.033$, with ABC Interventions (2.45 ± 0.17) having a higher level of planning horizon than Roanoke Controls (2.01 ± 0.07 , Bonferroni-corrected $p = 0.027$) and but not significantly different from ABC Controls (2.08 ± 0.20 , Bonferroni-corrected $p = 0.269$) (Fig. 4b). This result suggests that social decision-making in ABC Interventions, compared to Roanoke Controls, seems to be particularly influenced by future and long-term outcomes.”*

Discussion (Page 16) *“On the one hand, we observed that ABC Interventions planned further into the future in the MRT compared to Roanoke Controls... It would be important for future studies to examine the role of these underlying strategies (e.g., planning horizon and ToM) on social decision-making using experimental designs. It is notable that these effects found in the current study may be related to other behavioral outcomes in this and other study sample(s), such as their income and other economic outcomes and their personal-social well-being. Future work on this would add ecological validity of these process-oriented findings.”*

Discussion (Page 16) *“The high planning horizon of the ABC Interventions estimated from the MRT suggests their unique norm enforcement behavior during the UG may be motivated by such anticipated future personal costs but also potential social benefits. Consistent with this interpretation, behavior towards equality and fairness despite short-term cost to self has been proposed as important in stabilizing long-term cooperation^{7,8}. It has also been found that equality enforcement by punishing unequal offers is positively correlated with altruistic behaviors across cultures⁹.”*

Minor points:

The authors apply economic games to study social preferences. However, they do not follow the main rules developed within this field, which are non-deception and incentivization. I am aware of the differences among scientific fields, but I think the authors should at least discuss that they did not follow these rules and explain why.

Response: We thank the reviewer for pointing this out and we would like to explain our choices for the current study.

During the UG task, we told the participants that the offers, which were actually generated by an algorithm, were proposed by different *partners*, but we did not specify if the partners were human or a computer. We explicitly used this relatively vague term in order not to deceive the participants by saying they were playing a human being. Importantly, the pattern of rejection observed (across all groups) leads us to believe that participants considered “partners” as humans.

We chose to generate the offers based on an algorithm because we needed to have enough advantageous offers in order to test our hypotheses. Since people don't usually give high offers (i.e., offers advantageous to the Responder) when playing as the Proposer in the UG, we would not have been able to study the reaction of participants to different levels (especially advantageous ones) of inequality if we did not control this variable. The Trustee's responses in the MRT were generated using a k-nearest neighbors sampling algorithm on known responses from real players (similar to the practice in ¹³). Again, participants were told that they, as the Investor, were playing with another *partner* across the whole game, and the instruction was not specific about whether the repayments were sent by another person in real-time or based on previous responses from real players or from a computer. The practice here and in the UG was aimed at providing an ecological setting for the games (social tasks) while still controlling some aspects of the game.

Concerning the rule of incentivization, we provided monetary incentives (same for all groups) in both UG and MRT according to common practices used in these tasks. Participants were told that at the end of the UG one of the rounds in the UG would be randomly picked and they would be paid according to their decision on that round. In the MRT, participants were told that they would be paid based upon their earnings for the entire MRT task. In such ways, we tried to make sure different behaviors correspond to monetary differences in payments—higher gains in the games result in higher payments at the end—to motivate them to take the experiment seriously.

We have now addressed the issue of non-deception and incentivization in the Supplementary Note 6.

Supplementary Note 6 (Page 16 – 17 in Supplementary Information)
“Non-deception and incentivization

Participants were told that the offers in the UG task, which were actually generated by an algorithm, were proposed by different partners, but we did not specify if the partners were human or computer. We chose to generate the offers based on an algorithm because we needed to have enough advantageous offers in order to test our hypotheses. Since people don't usually give high offers (i.e., offers advantageous to the Responder) when playing as the Proposer in the UG, we would not have been able to study the reaction of participants to different levels (especially advantageous ones) of inequality if we did not control this variable. The Trustee's responses in the MRT were generated using a k-nearest neighbors sampling algorithm on known responses from real players (similar to the practice in ¹³). Again, participants were told that they, as the Investor, were playing with another partner across the whole game, and the instruction was not specific about whether the repayments were sent by another person in real-time or based on previous responses from real players or from a computer. The practice here and in the UG was aimed at providing an ecological setting for the games (social tasks) while still controlling some aspects of the game.

We provided monetary incentives (same for all groups) in both UG and MRT according to common practices used in these tasks. Participants were told that at the end of the UG

one of the rounds in the UG would be randomly picked and they would be paid according to their decision on that round. In the MRT, participants were told that they would be paid based upon their earnings for the entire MRT task. In such ways, we tried to make sure different behaviors correspond to monetary differences in payments—higher gains in the games result in higher payments at the end—to motivate them to take the experiment seriously.”

It would be helpful for the reader, if the authors would report exact p-values at least within a certain range such as $p < 0.001$ to $0.250 < p$, as p-values contain important information.

Response: We thank the reviewer’s suggestion about reporting exact p-values. All the exact p-values within the range of $p < 0.001$ to $p = 1.000$ are now reported in the manuscript and supplementary information.

I think it would be good to make the data and code for analysis publicly available on some online platform such as OSF.io. This facilitates data sharing.

Response: We thank the reviewer’s suggestion about data and code sharing. We will attempt to make the data and code available on an online platform. There are possible issues with our IRB. In the meantime we will make the data and code available by request.

References:

- 1 Bahry, D. L. & Wilson, R. K. Confusion or fairness in the field? Rejections in the ultimatum game under the strategy method. *Journal of Economic Behavior & Organization* **60**, 37-54 (2006).
- 2 Hennig-Schmidt, H., Li, Z.-Y. & Yang, C. Why people reject advantageous offers—Non-monotonic strategies in ultimatum bargaining: Evaluating a video experiment run in PR China. *Journal of Economic Behavior & Organization* **65**, 373-384 (2008).
- 3 Andreoni, J., Castillo, M. & Petrie, R. What do bargainers' preferences look like? Experiments with a convex ultimatum game. *American Economic Review* **93**, 672-685 (2003).
- 4 Huck, S. Responder behavior in ultimatum offer games with incomplete information. *Journal of Economic Psychology* **20**, 183-206 (1999).

- 5 Civai, C., Crescentini, C., Rustichini, A. & Rumiati, R. I. Equality versus self-interest in the brain: differential roles of anterior insula and medial prefrontal cortex. *Neuroimage* **62**, 102-112 (2012).
- 6 Tracer, D. Selfishness and fairness in economic and evolutionary perspective: An experimental economic study in Papua New Guinea. *Current Anthropology* **44**, 432-438 (2003).
- 7 Brosnan, S. F. & de Waal, F. B. Evolution of responses to (un) fairness. *Science* **346**, 1251776 (2014).
- 8 Cozzolino, P. J. Trust, cooperation, and equality: a psychological analysis of the formation of social capital. *British Journal of Social Psychology* **50**, 302-320 (2011).
- 9 Henrich, J. *et al.* Costly punishment across human societies. *Science* **312**, 1767-1770 (2006).
- 10 Fehr, E. & Fischbacher, U. Social norms and human cooperation. *Trends in cognitive sciences* **8**, 185-190 (2004).
- 11 Campbell, F. A. *et al.* Early childhood investments substantially boost adult health. *Science* **343**, 1478-1485 (2014).
- 12 Campbell, F. A. *et al.* Adult outcomes as a function of an early childhood educational program: an Abecedarian Project follow-up. *Developmental psychology* **48**, 1033 (2012).
- 13 King-Casas, B. *et al.* The rupture and repair of cooperation in borderline personality disorder. *science* **321**, 806-810 (2008).

Reviewers' Comments:

Reviewer #2:

Remarks to the Author:

Line 313 – delete “and”

Lines 309-323 – This section is still a bit confusing because there isn't a clear pattern among the three groups like there was with the UG, so it's harder for the reader to find the pattern. You go in to this in more detail in the Discussion, but as there isn't a clear pattern of either the ABC groups vs Roanoke or ABC Interventions vs the other two, it would be useful to clarify it here to help the reader out.

Line 368 – “note”, not “not”

Lines 401-203 – Don't both ABC groups show higher levels of planning into the future than the controls? So Line 402 should add “both ABC groups planned further into the future in the MRT compared to Roanoke Controls”. And then the ABC Controls group has a higher level of ToM than both ABC Interventions and the Roanoke Control, so line 404 should add “more mentalization steps than ABC Interventions and Roanoke Controls during this game.” And in line 405, which groups? You've been discussing three groups, so it's important to be clear that you mean the ABC groups.

Paragraph starting at line 418 – there is a lot in this paragraph. I would split it up into two paragraphs with the first addressing the UG and the second addressing how the apparently longer time horizons from the MRT may tell us something about advantageous inequity aversion in the UG. In particular, the added line at 438-440 is important, but doesn't fit where it is.

Line 425 – need a comma before which

Reviewer #3:

Remarks to the Author:

The manuscript has improved a lot and my previous remarks have all been addressed to my satisfaction. I do not have any further comments.

Response to Reviewers

We thank the reviewer and the editor for their suggestions and requests aimed at improving our manuscript. We have addressed all of them by providing point-by-point responses below. Changes in the manuscript have been made using the track changes feature of Microsoft Word.

REVIEWERS' COMMENTS:

Reviewer #2 (Remarks to the Author):

Line 313 – delete “and”

Response: We thank the reviewer for pointing this error out. It has been corrected accordingly.

(Last paragraph in Results) “... we found a significant main effect of group for planning horizon, $H(2) = 6.849$, $p = 0.033$, with ABC Interventions (2.45 ± 0.17) having a higher level of planning horizon than Roanoke Controls (2.01 ± 0.07 , Bonferroni-corrected $p = 0.027$) ~~and~~ but not significantly different from ABC Controls (2.08 ± 0.20 , Bonferroni-corrected $p = 0.269$) (Fig. 4b).”

Lines 309-323 – This section is still a bit confusing because there isn’t a clear pattern among the three groups like there was with the UG, so it’s harder for the reader to find the pattern. You go in to this in more detail in the Discussion, but as there isn’t a clear pattern of either the ABC groups vs Roanoke or ABC Interventions vs the other two, it would be useful to clarify it here to help the reader out.

Response: We appreciate the reviewer’s concern and suggestion. We have now revised this part by clarifying the group difference pattern at the beginning.

(Last paragraph in Results) “However, using model-based analysis, we were able to highlight group differences in decision-making strategies, with ABC Interventions having a higher level of planning horizon compared to Roanoke Controls and a lower level of ToM compared to ABC Controls. To be more specific...”

Line 368 – “note”, not “not”

Response: We thank the reviewer for pointing out this typo. We have corrected the manuscript accordingly.

(Paragraph 3 in Discussion) “We *note* that these differences are based on a relatively small sample which might raise the possibility of it being a false positive.”

Lines 401-203 – Don't both ABC groups show higher levels of planning into the future than the controls? So Line 402 should add "both ABC groups planned further into the future in the MRT compared to Roanoke Controls". And then the ABC Controls group has a higher level of ToM than both ABC Interventions and the Roanoke Control, so line 404 should add "more mentalization steps than ABC Interventions and Roanoke Controls during this game." And in line 405, which groups? You've been discussing three groups, so it's important to be clear that you mean the ABC groups.

Response: We appreciate the reviewer's suggestions and hence we have carefully rechecked the results for group comparison in planning horizon and theory of mind (ToM). Based on independent-samples Kruskal-Wallis H test on planning horizon and the follow-up two-tailed Mann-Whitney U tests, the ABC Control group (2.08 ± 0.20) does not have a higher level of planning horizon than the Roanoke Control group (2.01 ± 0.07 , uncorrected $p = 0.777$). Therefore, we state that "we observed that ABC Interventions planned further into the future in the MRT compared to Roanoke Controls". In addition, the ABC Control group (2.83 ± 0.24) does not have a higher level of ToM than the Roanoke Control group (2.62 ± 0.09 , uncorrected $p = 0.325$). Accordingly, we put that "the higher ToM parameter for ABC Controls indicated that this group utilized more mentalization steps than ABC Interventions during this game". Last, we agree with the reviewer that it is important to be clear about the groups we are referring to and have modified the text accordingly.

(Paragraph 5 in Discussion) *"These findings suggest that the similar overall behavior of the two ABC groups might be motivated by different social decision-making strategies."*

Paragraph starting at line 418 – there is a lot in this paragraph. I would split it up into two paragraphs with the first addressing the UG and the second addressing how the apparently longer time horizons from the MRT may tell us something about advantageous inequity aversion in the UG. In particular, the added line at 438-440 is important, but doesn't fit where it is.

Response: We thank the reviewer for the suggestions. We have split this paragraph into two with the first focusing on UG findings and the second discussing how the longer planning horizon suggested by our MRT results can bring insights into the inequality aversion results found in UG. Other than that, we have modified the latter paragraph to provide a more reasonable interpretation.

(Paragraph 7 in Discussion) *"The longer planning horizon of the ABC Interventions estimated from the MRT suggests their unique norm enforcement behavior during the UG may be motivated by such anticipated future personal costs but also potential social benefits. Consistent with this interpretation, equality enforcement by punishing unequal offers has been shown to be positively correlated with altruistic behaviors across cultures³⁹. Indeed, behavior towards equality and fairness despite short-term cost to self has been proposed as important in stabilizing long-term cooperation^{9,40}."*

Line 425 – need a comma before which

Response: We thank the reviewer for pointing this out and have revised accordingly.

(Paragraph 6 in Discussion) *“Importantly, studies²² including our own have also shown that this social norm is less enforced when self-interest comes into play, which may explain why it is rarely observed.”*